# Re-Examining Linear Embeddings for High-Dimensional Bayesian Optimization

**Benjamin Letham**
Facebook
Menlo Park, CA
bletham@fb.com

**Roberto Calandra**
Facebook AI Research
Menlo Park, CA
rcalandra@fb.com

**Akshara Rai**
Facebook AI Research
Menlo Park, CA
akshararai@fb.com

**Eytan Bakshy**
Facebook
Menlo Park, CA
ebakshy@fb.com

## Abstract

Bayesian optimization (BO) is a popular approach to optimize expensive-to-evaluate black-box functions. A significant challenge in BO is to scale to high-dimensional parameter spaces while retaining sample efficiency. A solution considered in existing literature is to embed the high-dimensional space in a lower-dimensional manifold, often via a random linear embedding. In this paper, we identify several crucial issues and misconceptions about the use of linear embeddings for BO. We study the properties of linear embeddings from the literature and show that some of the design choices in current approaches adversely impact their performance. We show empirically that properly addressing these issues significantly improves the efficacy of linear embeddings for BO on a range of problems, including learning a gait policy for robot locomotion.

## 1   Introduction

Bayesian optimization (BO) is a robust, sample-efficient technique for optimizing expensive-to-evaluate black-box functions [34, 24]. BO has been successfully applied to diverse applications, ranging from automated machine learning [44, 22] to robotics [32, 6, 40]. One of the most active topics of research in BO is how to extend current methods to higher-dimensional spaces. A common framework to tackle this problem is to consider a high-dimensional BO (HDBO) task as a standard BO problem in a low-dimensional embedding, where the embedding can be either linear (typically a random projection) or nonlinear (*e.g.*, via a multi-layer neural network); see Sec. 2 for a full review. An advantage of this framework is that it explicitly decouples the problem of finding low-dimensional representations suitable for optimization from the actual optimization technique.

In this paper we study the use of linear embeddings for HDBO, and in particular we re-examine prior efforts to use random linear projections. Random projections are attractive because, by the Johnson-Lindenstrauss lemma, they can be approximately distance-preserving [23] without requiring data to learn the embedding. Random embeddings come with strong theoretical guarantees, but have shown mixed empirical performance for HDBO. Our goal here is not just to present a new HDBO method, but rather to improve understanding of important considerations for BO in an embedding.

The contributions of this paper are: **1)** We provide new results that identify why linear embeddings have performed poorly in HDBO. We show that existing approaches can produce representations that are not well-modeled by a Gaussian process (GP), or do not contain an optimum (Sec. 4). **2)** We construct a representation with better properties for BO (Sec. 5): modelability is improved with a Mahalanobis kernel tailored for linear embeddings and by adding polytope bounds to the embedding, and we show how to maintain a high probability that the embedding contains an optimum. **3)** We combine these improvements to form a new linear-embedding HDBO method, ALEBO, and show empirically that it outperforms a wide range of HDBO techniques, including on test functions up to $D$=1000, with black-box constraints, and for gait optimization of a multi-legged robot (Secs. 6

and 7). These results show empirically that we have identified several important elements impacting the BO performance of linear embedding methods. Code to reproduce the results of this paper is available at `github.com/facebookresearch/alebo`.

## 2    Related Work

There are generally two approaches to extending BO into high dimensions. The first is to produce a low-dimensional embedding, do standard BO in this low-dimensional space, and then project up to the original space for function evaluations. The foundational work on embeddings for BO is REMBO [49], which uses a random projection matrix. Sec. 3 provides a thorough description of REMBO and several subsequent approaches based on random linear embeddings [39, 37, 5]. If derivatives of $f$ are available, the active subspace method can be used to recover a linear embedding [7, 11], or approximate gradients can be used [10]. A linear embedding can also be learned end-to-end with the GP [15], however this requires estimating $D \times d$ parameters, which is infeasible in settings where the total number of evaluations is much less than $D$. BO can also be done in nonlinear embeddings through VAEs [17, 33, 35]. An attractive aspect of random embeddings is that they can be extremely sample-efficient, since the only model to be estimated is a low-dimensional GP.

The second approach to extend BO to high dimensions is to make use of surrogate models that better handle high dimensions, typically by imposing additional structure on the problem. Work along these lines include GPs with an additive kernel [26, 50, 13, 51, 42, 36], cylindrical kernels [38], or deep neural network kernels [1]. Random forests are used as the surrogate model in SMAC [22].

Here, we focus on the embedding approach, and in particular the use of linear embeddings for HDBO. While REMBO can perform well in some HDBO tasks, subsequent papers have found it can perform poorly even on synthetic tasks with a true low-dimensional linear subspace [*e.g.,* 37]. In this paper, we analyze the properties of linear embeddings as they relate to BO, and show how to improve the representation of the function we seek to optimize.

## 3    Problem Framework and REMBO

In this section, we define the problem framework and notation, and then describe REMBO, along with known challenges and follow-up work that has been proposed to address those issues.

**Bayesian Optimization**     We consider the problem $\min_{\boldsymbol{x} \in \mathcal{B}} f(\boldsymbol{x})$ where $f$ is a black-box function and $\mathcal{B}$ are box bounds. We assume gradients of $f$ are unavailable. The box bounds on $\boldsymbol{x}$ specify the range of values that are reasonable or physically possible to evaluate. For instance, [18] used BO for an environmental remediation problem in which each $x_i$ represents a pumping rate of a particular pump, which has physical limitations. The problem may also include nonlinear constraints $c_j(\boldsymbol{x}) \leq 0$ where each $c_j$ is itself a black-box function. BO is a form of sequential model-based optimization, where we fit a surrogate model for $f$ that is used to identify which parameters $\boldsymbol{x}$ should be evaluated next. The surrogate model is typically a GP, $f \sim \mathcal{GP}(m(\cdot), k(\cdot, \cdot))$, with mean function $m(\cdot)$ and a kernel $k(\cdot, \cdot)$. Under the GP prior, the posterior for the value of $f(\boldsymbol{x})$ at any point in the space is a normal distribution with closed-form mean and variance. Using that posterior, we construct an acquisition function $\alpha(\boldsymbol{x})$ that specifies the utility of evaluating $f$ at $\boldsymbol{x}$, such as Expected Improvement (EI) [25]. We find $\boldsymbol{x}^* \in \arg\max_{\boldsymbol{x} \in \mathcal{B}} \alpha(\boldsymbol{x})$, and in the next iteration evaluate $f(\boldsymbol{x}^*)$.

GPs are useful for BO because they provide a well-calibrated posterior in closed form. With many kernels and acquisition functions, $\alpha(\boldsymbol{x})$ is differentiable and can be efficiently optimized. However, typical kernels like the ARD RBF kernel have significant limitations. GPs are known to predict poorly for dimension $D$ larger than 15–20 [49, 31, 37], which prevents the use of standard BO in high dimensions. In HDBO, the objective $f : \mathbb{R}^D \rightarrow \mathbb{R}$ operates in a high-dimensional ($D$) space, which we call the *ambient space*. When using linear embeddings for HDBO, we assume there exists a low-dimensional linear subspace that captures all of the variation of $f$. Specifically, let $f_d : \mathbb{R}^d \rightarrow \mathbb{R}$, $d \ll D$, and let $\boldsymbol{T} \in \mathbb{R}^{d \times D}$ be a projection from $D$ down to $d$ dimensions. The linear embedding assumption is that $f(\boldsymbol{x}) = f_d(\boldsymbol{T}\boldsymbol{x}) \ \forall \boldsymbol{x} \in \mathbb{R}^D$. $\boldsymbol{T}$ is unknown, and we only have access to $f$, not $f_d$. We assume, without any loss of generality, that the box bounds are $\mathcal{B} = [-1, 1]^D$; the ambient space can always be scaled to these bounds.

**Bayesian Optimization via Random Embeddings** REMBO [49] specifies a $d_e$-dimensional embedding via a random projection matrix $\boldsymbol{A} \in \mathbb{R}^{D \times d_e}$ with each element i.i.d. $\mathcal{N}(0, 1)$. BO is done in the embedding to identify a point $\boldsymbol{y} \in \mathbb{R}^{d_e}$ to be evaluated, which is given objective value $f(\boldsymbol{Ay})$. Without box bounds, REMBO comes with a strong guarantee: if $d_e \geq d$, then with probability 1 the embedding contains an optimum [49, Thm. 2]. Unfortunately, things become complicated when there are box bounds (or any other sort of bound) in the ambient space. One may select a point $\boldsymbol{y}$ in the embedding to be evaluated and find that its projection to the ambient space, $\boldsymbol{Ay}$, falls outside $\mathcal{B}$. The embedding subspace is guaranteed to contain an optimum to the box-bounded problem [49, Thm. 3], but that optimum is *not* guaranteed to project up inside $\mathcal{B}$. When function evaluations are restricted to the box bounds, there is no guarantee that we can find an optimum in the embedding.

REMBO introduces three heuristics for handling box bounds. First, the embedding is given box bounds $[-\sqrt{d_e}, \sqrt{d_e}]^{d_e}$. Second, if a point $\boldsymbol{y}$ in the embedding projects up outside $\mathcal{B}$, then it is clipped to $\mathcal{B}$. Let $p_{\mathcal{B}} : \mathbb{R}^D \to \mathbb{R}^D$ be the $L^2$ projection that maps $\boldsymbol{x}$ to its nearest point in $\mathcal{B}$. Then, $\boldsymbol{y}$ is given value $f(p_{\mathcal{B}}(\boldsymbol{Ay}))$, which can always be evaluated. Note that clipping to $\mathcal{B}$ renders the projection of $\boldsymbol{y}$ a nonlinear transformation whenever $\boldsymbol{Ay} \notin \mathcal{B}$. Third, the optimization is done with $k$=4 separate projections, to improve the chances of generating an embedding that contains an optimum inside $[-\sqrt{d_e}, \sqrt{d_e}]^{d_e}$. No data can be shared across the embeddings, which reduces sample efficiency.

**Extensions of REMBO** [4] considers the issue of non-injectivity, where the $L^2$ projection causes many points in the embedding to map to the same vertex of $\mathcal{B}$. They introduce REMBO-$\phi k_\Psi$, which uses a warped kernel to reduce non-injectivity. [5] defines a projection matrix $\boldsymbol{B} \in \mathbb{R}^{d \times D}$ that maps from the ambient space down to the embedding, and replaces the $L^2$ projection entirely with a projection $\gamma$ that maps $\boldsymbol{y}$ to the closest point in $\mathcal{B}$ that satisfies $\boldsymbol{Bx} = \boldsymbol{y}$. The $\gamma$ projection eliminates the need for heuristic box bounds on the embedding, while mapping to the same set of points as the $L^2$ projection. This method is called REMBO-$\gamma k_\Psi$. [3] studies the projection matrix and shows that BO performance can be improved for small $d$ by sampling each row of $\boldsymbol{A}$ from the unit hypersphere $\mathbb{S}^{d_e-1}$. If $\boldsymbol{z} \sim \mathcal{N}(\boldsymbol{0}, \boldsymbol{I}_{d_e})$, then $\frac{\boldsymbol{z}}{||\boldsymbol{z}||}$ is a sample from $\mathbb{S}^{d_e-1}$, so this amounts to normalizing the rows of the usual REMBO projection matrix. HeSBO [37] avoids clipping to $\mathcal{B}$ by changing the projection matrix $\boldsymbol{A}$ so that each row of $\boldsymbol{A}$ has a single non-zero element in a random column, which is randomly set to $\pm 1$. In the ambient space, $x_i = \pm y_j$, where $j \sim \text{Unif}\{1, d_e\}$, $\pm$ is chosen uniformly at random, and $\boldsymbol{y} \in [-1, 1]^{d_e}$.

## 4 Challenges with Linear Embeddings

The heuristics just described introduce several issues that impact HDBO performance of linear embedding methods. We highlight one recent observation from [5], that most points in the embedding project up outside $\mathcal{B}$, and discuss three novel observations on why existing methods can struggle to learn useful high-dimensional surrogates.

**Projection to the facets of $\mathcal{B}$ produces a nonlinear distortion in the function.** The function value at any point in the embedding is evaluated as $f(p_{\mathcal{B}}(\boldsymbol{Ay}))$. For points $\boldsymbol{y}$ that project up outside of $\mathcal{B}$, this will be a nonlinear mapping despite the use of a linear embedding. This has a powerful, detrimental effect on the ability to model $f$ in the embedding. Fig. 1 provides visualizations of an actual REMBO embedding for two classic test functions, both extended to $D$=100 by adding unused variables. The embedding for the Branin function contains all three optima, however there is clear, nonlinear distortion to the function caused by the clipping to $\mathcal{B}$. The embedding for the Hartmann6 function is even more heavily distorted. The distortion induced by clipping to a facet depends on the relative angles of the facet and the true embedding. Projection to a facet essentially induces a non-stationarity in the kernel: each of the $2D$ facets sits at different angles to the true subspace, and so the change in the rate of function variance will differ for each. To correct for the non-stationarity, we would have to estimate the true subspace $\boldsymbol{T}$, which with $d \times D$ entries is not feasible for $D$ large.

The appeal of embeddings for HDBO is that they enable the use of standard BO on the embedding. However, these results show that for the REMBO projection with box bounds we may not be able to model in the embedding with a standard (stationary) GP, even if the function is well-modeled in the true low-dimensional space. The problem is especially acute for $d_e > 2$ where, as we will see next, nearly all points in the embedding map to one of the $2D$ facets.

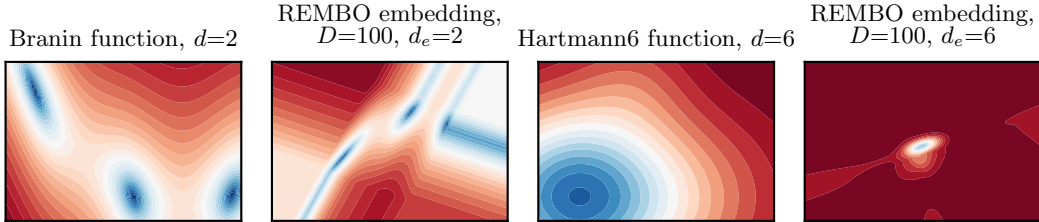

| Branin function, $d$=2 | REMBO embedding, $D$=100, $d_e$=2 | Hartmann6 function, $d$=6 | REMBO embedding, $D$=100, $d_e$=6 |

Figure 1: A visualization of REMBO embeddings for two test functions. *(Far left)* The Branin function, $d$=2, extended to $D$=100. *(Center left)* A REMBO embedding of the $D$=100 Branin function. *(Center right)* A center slice of the $d$=6 Hartmann6 function, similarly extended to $D$=100. *(Far right)* The same slice of a REMBO embedding of that function. The embedding produces distortions and non-stationarity in the function that render it difficult to model.

**Most points in the embedding map to the facets of $\mathcal{B}$.** Fig. 2 shows the probability that an interior point in the embedding projects up to the interior of $\mathcal{B}$, measured empirically (with 1000 samples) by sampling $\boldsymbol{y}$ uniformly from $[-\sqrt{d_e}, \sqrt{d_e}]^{d_e}$, sampling $\boldsymbol{A}$ with $\mathcal{N}(0,1)$ entries, and then checking if $\boldsymbol{Ay} \in \mathcal{B}$. Even for small $D$, with $d_e > 2$ practically all of the volume in the embedding projects up outside the box bounds, and is thus clipped to a facet of $\mathcal{B}$. This is an issue because it means the optimization will be done primarily on the facets of $\mathcal{B}$. We saw in Fig. 1 that the function behaves very differently on points projected to the facets and that the function is non-stationary in these parts of the space. The problem cannot be resolved by simply shrinking the box bounds in the embedding. [5] provides an excellent study of this issue and shows that with the REMBO strategy there is no good way to set box bounds in the embedding.

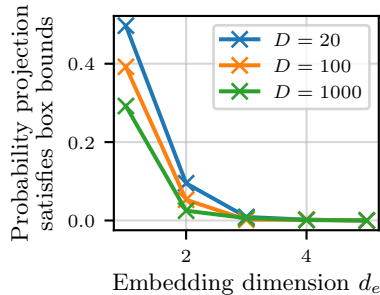

Figure 2: The probability a randomly selected point in the REMBO embedding satisfies the ambient box bounds after being projected up. For $d_e > 2$, nearly all points in the embedding project outside the bounds.

**Linear projections do not preserve product kernels.** Although less visible than that produced by the projection to the facets, there is also distortion to interior points just from the linear projection $\boldsymbol{A}$. The ARD kernels typically used in GP modeling are product kernels that decompose the covariance into that across each dimension. Inside the embedding, moving along a single dimension will move across all dimensions of the ambient space, at rates depending on the projection matrix. Thus a product kernel in the true subspace will not produce a product kernel in the embedding; this is shown mathematically in Proposition 1.

**Linear embeddings can have a low probability of containing an optimum.** HeSBO avoids the challenges of REMBO related to box bounds: all interior points in the embedding map to interior points of $\mathcal{B}$, and there is no need for the $L^2$ projection and so the ability to model in the embedding is improved. However, for $d_e > 1$ the embedding is not guaranteed to contain an optimum with high probability, and in fact the probability of containing an optimum can be low. Consider the example of an axis-aligned true subspace: $f$ operates only on $d$ elements of $\boldsymbol{x}$, denoted $\mathcal{I} = \{i_1, \ldots, i_d\}$. For $d = 2$ and $d_e \geq 2$, there are three possible HeSBO embeddings: $x_{i_1}$ and $x_{i_2}$ map to different features in the embedding, $x_{i_1} = x_{i_2}$, or $x_{i_1} = -x_{i_2}$. These three embeddings are visualized in the supplement in Sec. S1. In the first case the embedding successfully captures the entire true subspace and we can expect the optimization to be successful. However, in the other two cases the embedding is only able to reach the diagonals of the true subspace, which will not reach the optimal value, unless $f$ happens to have an optimum on the diagonal. Under a uniform prior on the location of optima, we can compute analytically the probability that the HeSBO embedding contains an optimum (see Sec. S1). For instance, with $d = 6$, $d_e = 12$ gives only a 22% chance of recovering an optimum. Relative to REMBO, HeSBO improves the ability to model and optimize in the embedding, but reduces the chance of the embedding containing an optimum. Empirically, this trade-off leads to HeSBO often having better HDBO performance than REMBO. Here we also wish to improve our ability to model and optimize in the embedding, which we will show can be done while maintaining a higher chance of the embedding containing an optimum, further improving HDBO performance.

# 5 Learning and Optimizing in Linear Embeddings

We now describe how the embedding issues described in Sec. 4 can be overcome. Similarly to [5], we define the embedding via a matrix $\boldsymbol{B} \in \mathbb{R}^{d_e \times D}$ that projects from the ambient space down to the embedding, and $f_B(\boldsymbol{y}) = f(\boldsymbol{B}^\dagger \boldsymbol{y})$ as the function evaluated on the embedding, where $\boldsymbol{B}^\dagger$ denotes the matrix pseudo-inverse. The techniques we develop here are applicable to any linear embedding, not just random embeddings.

**A Kernel for Learning in a Linear Embedding** As discussed in Sec. 4, a product kernel over dimensions of the true subspace (*e.g.*, ARD) does not translate to a product kernel over the embedding. This result gives the appropriate kernel structure.

**Proposition 1.** *Suppose the function on the true subspace is drawn from a GP with an ARD RBF kernel: $f_d \sim \mathcal{GP}(m(\cdot), k_{RBF}(\cdot, \cdot))$. For any pair of points in the embedding $\boldsymbol{y}$ and $\boldsymbol{y}'$,*

$$Cov[f_B(\boldsymbol{y}), f_B(\boldsymbol{y}')] = \sigma^2 \exp\left(-(\boldsymbol{y} - \boldsymbol{y}')^\top \boldsymbol{\Gamma} (\boldsymbol{y} - \boldsymbol{y}')\right) ,$$

*where $\sigma^2$ is the kernel variance of $f_d$, and $\boldsymbol{\Gamma} \in \mathbb{R}^{d_e \times d_e}$ is symmetric and positive definite.*

*Proof.* To determine the covariance function in the embedding, we first project up to the ambient space and then project down to the true subspace: $f_B(\boldsymbol{y}) = f(\boldsymbol{B}^\dagger \boldsymbol{y}) = f_d(\boldsymbol{T}\boldsymbol{B}^\dagger \boldsymbol{y})$. Then,

$$\begin{aligned}
\mathrm{Cov}[f_B(\boldsymbol{y}), f_B(\boldsymbol{y}')] &= \mathrm{Cov}[f_d(\boldsymbol{T}\boldsymbol{B}^\dagger \boldsymbol{y}), f_d(\boldsymbol{T}\boldsymbol{B}^\dagger \boldsymbol{y}')] \\
&= \sigma^2 \exp\left(-(\boldsymbol{T}\boldsymbol{B}^\dagger \boldsymbol{y} - \boldsymbol{T}\boldsymbol{B}^\dagger \boldsymbol{y}')^\top \boldsymbol{D} (\boldsymbol{T}\boldsymbol{B}^\dagger \boldsymbol{y} - \boldsymbol{T}\boldsymbol{B}^\dagger \boldsymbol{y}')\right) ,
\end{aligned}$$

where $\boldsymbol{D} = \mathrm{diag}\left(\left[\frac{1}{2\ell_1^2}, \ldots, \frac{1}{2\ell_d^2}\right]\right)$ are the inverse lengthscales. Let $\boldsymbol{\Gamma} = (\boldsymbol{T}\boldsymbol{B}^\dagger)^\top \boldsymbol{D} (\boldsymbol{T}\boldsymbol{B}^\dagger)$. Because $\boldsymbol{D}$ is positive definite, $\boldsymbol{\Gamma}$ is symmetric and positive definite. □

This kernel replaces the ARD Euclidean distance with a Mahalanobis distance, and so we refer to it as the Mahalanobis kernel. A similar result was found by [15], which showed that an RBF kernel in a linear subspace implies a Mahalanobis kernel in the high-dimensional, ambient space. Similar kernels have also been used for GP regression in other settings [48, 43].

Prop. 1 shows that the impact of the linear projection on the kernel can be correctly handled by fitting a $\frac{d_e(d_e+1)}{2}$-parameter distance metric rather than the typical $d_e$-parameter ARD metric. We handle uncertainty in $\boldsymbol{\Gamma}$ by posterior sampling from a Laplace approximation of its posterior; this is described in Sec. S2 in the supplement. The use of this kernel is vital for obtaining good model fits in the embedding, as shown in Fig. 3. For Fig. 3, a 6-d random linear embedding was generated for the Hartmann6 $D$=100 problem, and 100 training and 50 test points were randomly sampled that mapped to the interior of $\mathcal{B}$ (so, they have no distortion from clipping). The ARD RBF kernel entirely failed to learn the function on the embedding and simply predicted the mean; the Mahalanobis kernel made accurate out-of-sample predictions. Further details are given in Sec. S2, along with learning curves.

A similar argument as Prop. 1 shows that with linear embeddings, stationarity in the true subspace implies stationarity in the embedding; see Sec. S3. As discussed there, this result does not hold with clipping to box bounds, which effectively produces a nonlinear embedding.

**Avoiding Nonlinear Projections** The most significant distortions seen in Fig. 1 result from clipping projected points to $\mathcal{B}$. We can avoid this by constraining the optimization in the embedding to points that do not project up outside the bounds, that is, $\boldsymbol{B}^\dagger \boldsymbol{y} \in \mathcal{B}$. Let $\alpha(\boldsymbol{y})$ be the acquisition function evaluated in the embedding. We select the next point to evaluate by solving

$$\max_{\boldsymbol{y} \in \mathbb{R}^{d_e}} \alpha(\boldsymbol{y}) \quad \text{subject to} \quad -\mathbf{1} \le \boldsymbol{B}^\dagger \boldsymbol{y} \le \mathbf{1} . \tag{1}$$

Box bounds on $\boldsymbol{y}$ are not required. The constraints $-\mathbf{1} \le \boldsymbol{B}^\dagger \boldsymbol{y} \le \mathbf{1}$ are all linear, so they form a polytope and can be handled with off-the-shelf optimization tools; we use Scipy's SLSQP. Sec. S4 in the supplement provides visualizations of the embedding subject to these constraints. Within this space, the projection is entirely linear and can be effectively modeled with the Mahalanobis kernel.

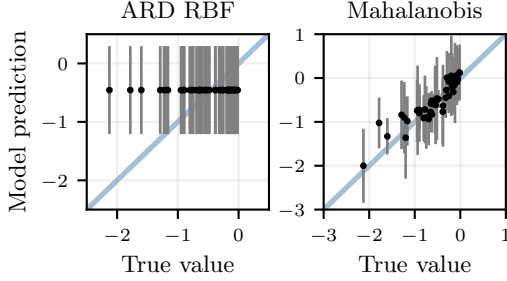

Figure 3: Predictions (mean, and in error bars two standard deviations, of the posterior predictive distribution) on a test set of 50 points from a 6-d embedding of the Hartmann6 $D$=100 problem, with models fit to 100 training points. The ARD RBF kernel is unable to learn in the embedding and predicts the mean. The Mahalanobis kernel makes accurate test-set predictions.

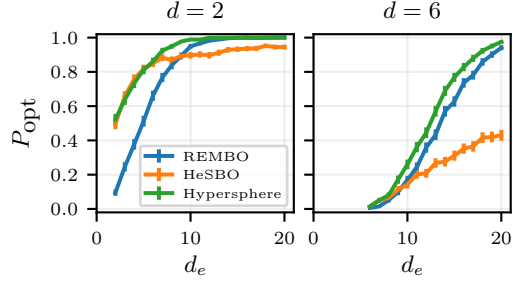

Figure 4: Probability the embedding contains an optimum ($P_{\text{opt}}$) when restricted to the constraints of (1), under a uniform prior for the location of the optima and $D$=100, for three embedding strategies. Setting $d_e > d$ rapidly increases $P_{\text{opt}}$, and high probabilities can achieved with reasonable values of $d_e$. Hypersphere sampling produces the best embedding, particularly for $d$ small.

**The Probability the Embedding Contains an Optimum**  Restricting the embedding with the constraints in (1) eliminates distortions from clipping to $\mathcal{B}$, but it also reduces the volume of the ambient space that can be reached from the embedding and thus reduces the probability that the embedding contains an optimum. To understand the performance of BO in the linear embedding, it is critical to understand this probability, which we denote $P_{\text{opt}}$. Recall that with clipping, the REMBO theoretical result does not hold when function evaluations are restricted to box bounds, and so even REMBO generally has $P_{\text{opt}} < 1$. We now describe how $P_{\text{opt}}$ can be estimated, and increased.

$P_{\text{opt}}$ depends on where optima are in the ambient space—for instance, an optimum at $\mathbf{0}$ is always contained in the embedding. Suppose the true subspace has an optimum at $\mathbf{z}^*$. Then, $\mathcal{O}(\mathbf{T}, \mathbf{z}^*) = \{\mathbf{x} : \mathbf{T}\mathbf{x} = \mathbf{z}^*\}$ is the set of optima in the ambient space. We wish to determine if these can be reached from the embedding. The points $\mathbf{x}$ that can be reached from the embedding are those for which there exists a $\mathbf{y}$ in the embedding that projects up to $\mathbf{x}$, that is, $\mathbf{B}^\dagger \mathbf{y} = \mathbf{x}$. Since the embedding is produced from $\mathbf{B}\mathbf{x}$, the points that can be reached from the embedding are $\mathcal{E}(\mathbf{B}) = \{\mathbf{x} : \mathbf{B}^\dagger \mathbf{B}\mathbf{x} = \mathbf{x}\}$. The embedding contains an optimum if and only if the intersection $\mathcal{O}(\mathbf{T}, \mathbf{z}^*) \cap \mathcal{E}(\mathbf{B}) \cap \mathcal{B}$ is non-empty. Given a prior for the locations of optima (that is, over $\mathbf{T}$ and $\mathbf{z}^*$),

$$P_{\text{opt}} = \mathbb{E}_{\mathbf{B}, \mathbf{T}, \mathbf{z}^*} \left[ \mathbf{1}_{\mathcal{O}(\mathbf{T}, \mathbf{z}^*) \cap \mathcal{E}(\mathbf{B}) \cap \mathcal{B} \neq \varnothing} \right] . \tag{2}$$

Importantly, $\mathcal{O}(\mathbf{T}, \mathbf{z}^*)$, $\mathcal{E}(\mathbf{B})$, and $\mathcal{B}$ are all polyhedra, so their intersection can be tested by solving a linear program (see Sec. S5 in the supplement). The expectation can be estimated with Monte Carlo sampling from the prior over $\mathbf{T}$ and $\mathbf{z}^*$, and from the chosen generating distribution of $\mathbf{B}$.

For our analysis here, we give $\mathbf{T}$ a uniform prior over axis-aligned subspaces as described in Sec. 4, and we give $\mathbf{z}^*$ a uniform prior in that subspace. Under these uniform priors, we can evaluate (2) to compute $P_{\text{opt}}$ as a function of $\mathbf{B}$, $D$, $d$, and $d_e$. Fig. 4 shows these probabilities for $D$=100 as a function of $d$ and $d_e$, with three strategies for generating the projection matrix: the REMBO strategy of $\mathcal{N}(0,1)$, the HeSBO projection matrix, and the unit hypersphere sampling described in Sec. 3. Increasing $d_e$ above $d$ rapidly improves the probability of containing an optimum. For $d = 6$, with $d_e = 6$ $P_{\text{opt}}$ is nearly 0, while increasing $d_e$ to 12 is sufficient to raise $P_{\text{opt}}$ to 0.5 and with $d_e = 20$ it is nearly 1. Across all values of $d$ and $d_e$, hypersphere sampling produces the embedding with the best chance of containing an optimum. Sec. S5 shows $P_{\text{opt}}$ for more values of $D$ and $d$. Hypersphere sampling and selecting $d_e > d$ are important techniques for maintaining a high $P_{\text{opt}}$.

**A New Method for BO with Linear Embeddings**  We combine the results and insight gained above into a new method for HDBO, called adaptive linear embedding BO (ALEBO), since the kernel metric and embedding bounds adapt with $\mathbf{B}$. It is summarized in Algorithm 1.

**Algorithm 1:** ALEBO for linear embedding BO.

**Data:** $D$, $d_e$, $n_{\text{init}}$, $n_{\text{BO}}$.
**Result:** Approximate optimizer $\boldsymbol{x}^*$.

1 Generate a random projection matrix $\boldsymbol{B}$ by sampling $D$ points from the hypersphere $\mathbb{S}^{d_e - 1}$.
2 Generate $n_{\text{init}}$ random points $\boldsymbol{y}^i$ in the embedding using rejection sampling to satisfy the constraints of (1).
3 Let $\mathcal{D} = \{(\boldsymbol{y}^i, f(\boldsymbol{B}^\dagger \boldsymbol{y}^i))\}_{i=1}^{n_{\text{init}}}$ be the initial data.
4 **for** $j = 1, \dots, n_{BO}$ **do**
5     Fit a GP to $\mathcal{D}$ with the Mahalanobis kernel, using posterior sampling (Sec. S2).
6     Use the GP to find $\boldsymbol{y}^j$ that maximizes the acquisition function according to (1).
7     Update $\mathcal{D}$ with $(\boldsymbol{y}^j, f(\boldsymbol{B}^\dagger \boldsymbol{y}^j))$.
8 **return** $\boldsymbol{B}^\dagger \boldsymbol{y}^*$, *for the best point* $\boldsymbol{y}^*$.

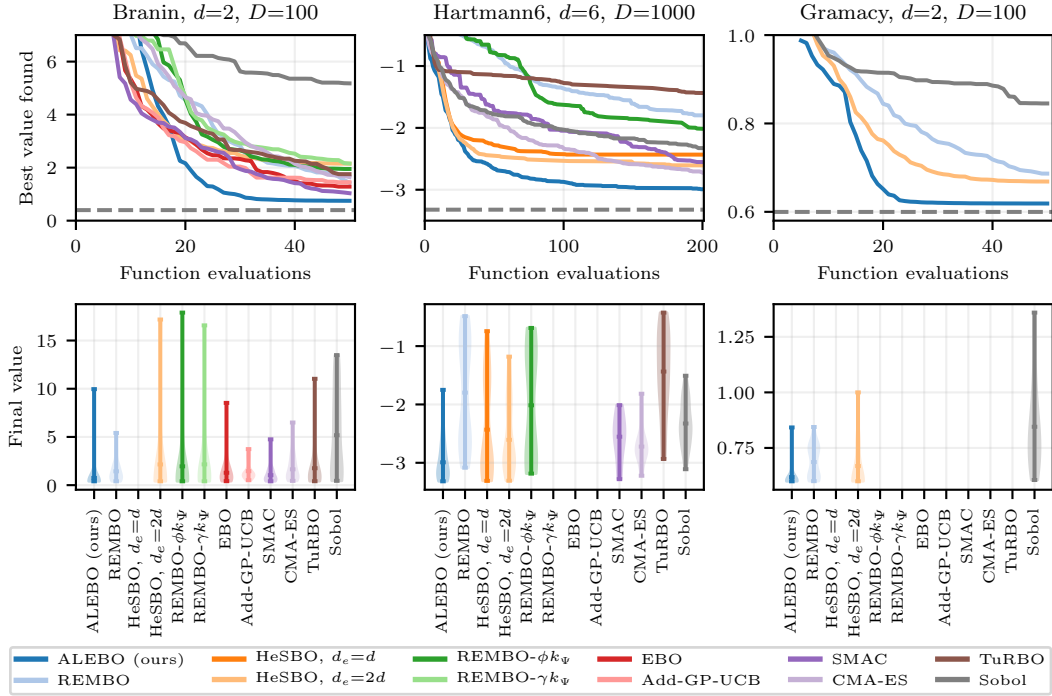

Figure 5: Optimization (minimization) performance on three HDBO benchmarks. (*Top*) Best value by each iteration, averaged over repeated runs. (*Bottom*) Distribution of the final best value. For all three tasks, ALEBO achieved the best average performance, with low variance.

## 6 Benchmark Experiments

Past work has shown REMBO can perform poorly even on problems that have a true, linear subspace, despite this being the setting that REMBO should be best-suited for. A major source of this poor performance are the modeling issues described in Sec. 4, which we demonstrate by showing that with the new developments in Sec. 5, ALEBO can achieve state-of-the-art performance on this class of problems (those with a true linear subspace). We compare performance to a broad selection of existing methods: REMBO; REMBO variants $\phi k_\Psi$, $\gamma k_\Psi$, and HeSBO; additive kernel methods Add-GP-UCB [26] and Ensemble BO (EBO) [51]; SMAC; CMA-ES, an evolutionary strategy [19]; TuRBO, trust region BO [12]; and quasirandom search (Sobol). Sec. S8 additionally compares to LineBO [27]. For ALEBO we took $d_e = 2d$ for these experiments. In their evaluation of HeSBO, [37] used $d_e = 2d$ for $d = 2$ but $d_e = d$ on the Hartmann6 problem; we evaluate both choices.

Fig. 5 shows optimization performance for three HDBO tasks, averaged over 50 runs: Branin extended to $D$=100, Hartmann6 extended to $D$=1000, and Gramacy extended to $D$=100. The Gramacy problem

[18] includes two black-box constraints, which are naturally handled by linear embedding methods (see Sec. S7; an advantage of linear embedding HDBO is that it is agnostic to the acquisition function). For the $D$=1000 problem, REMBO-$\gamma k_\Psi$, EBO, and Add-GP-UCB did not finish a single run after 24 hours and so were terminated. These methods, along with TuRBO, SMAC, and CMA-ES, also do not support blackbox constraints and so were not used for Gramacy. Sec. S8 provides additional details, additional experimental results (plots of log regret and error bars), two additional benchmark problems (including a non-axis-aligned problem), and an extended discussion of the results.

Consistent with past studies, REMBO performance was variable, and in the $D$=1000 problem it performed worse than random. With the adjustments in Sec. 5, ALEBO significantly improved HDBO performance relative to other linear embedding methods, and achieved the best average optimization performance overall. ALEBO also had low variance in the final best-value, which is important in real applications where one can typically only run one optimization run. These results show that with the adjustments in ALEBO, linear embedding BO is the best-performing method on linear subspace benchmark problems, as it ought to be. Sec. S8 gives a sensitivity analysis with respect to $D$ and $d_e$ (robust for $d_e > d$), and an ablation study of the different components of ALEBO.

## 7   Real-World Problems

**Constrained Neural Architecture Search**
We evaluated ALEBO performance on constrained neural architecture search (NAS) for convolutional neural networks using models from NAS-Bench-101 [53]. The NAS problem was to design a cell topology defined by a DAG with 7 nodes and up to 9 edges, which includes designs like ResNet [20] and Inception [47]. We created a $D = 36$ parameterization, producing a HDBO problem. The objective was to maximize CIFAR-10 test-set accuracy, subject to a constraint that training time was less than 30 mins; see Sec. S9 for full details. Sample efficiency is critical for NAS due to long training times on GPUs or TPUs. Fig. 6 shows optimization performance on this problem. ALEBO significantly improved accuracy relative to the other linear embedding approaches, which did not improve over Sobol, and was a best-performing method.

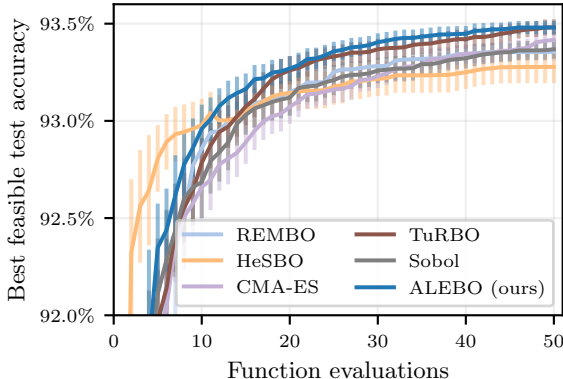

Figure 6: Best-feasible CIFAR10 test-set accuracy by each iteration for the $D = 36$ constrained NAS problem, showing mean and two standard errors over 100 repeated runs. ALEBO was a best-performing method, and the only embedding method that outperformed random search.

**Policy Search for Robot Locomotion**   We applied ALEBO to the problem of learning walking controllers for a simulated hexapod robot. Sample efficiency is crucial in robotics as collecting data on real robots is time consuming and can cause wear-and-tear on the robot. We optimized the walking gait of the "Daisy" robot [21], which has 6 legs with 3 motors in each leg, and was simulated in PyBullet [8]. The goal was to learn policy parameters that enable the robot to walk to a target location while avoiding high joint velocities and height deviations; details are given in Sec. S10. We use a Central Pattern Generator (CPG) [9] with $D = 72$ to control the robot. The 72-dimensional controller assumes each joint is independent of the others; a lower-dimensional embedding can be constructed by coupling multiple joints. For example, the tripod gait in hexapods assumes three sets of legs synced and out of phase with the remaining three legs, which produces an 11-dimensional parameterization. The existence of such low-dimensional parameterizations motivates the use of embedding methods for this problem, though there is no known *linear* low-dimensional representation.

Fig. 7 shows optimization performance on this task. ALEBO performed the best of the linear embedding methods, and also outperformed EBO, SMAC, and Sobol. REMBO performed poorly on this problem, only slightly better than random. The ALEBO results show that REMBO did not do poorly because linear embedding methods cannot learn on this problem, rather it was because of the issues described and corrected in this paper. CMA-ES outperformed all of the HDBO methods.

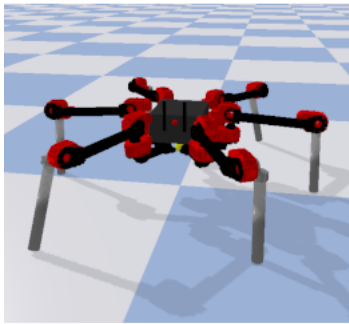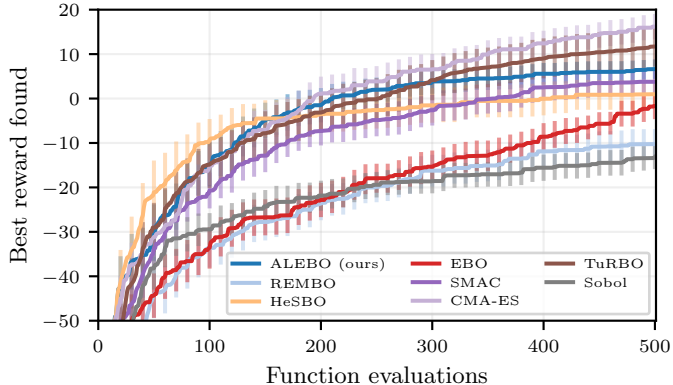

Figure 7: (*Left*) The simulated hexapod robot Daisy. (*Right*) Optimization (maximization) performance on the $D = 72$ locomotion task, showing mean and two standard errors over 100 runs. ALEBO significantly improved over REMBO, but all HDBO methods were outperformed by CMA-ES.

CMA-ES is model free, suggesting that the underlying models in the HDBO methods are not well suited for this particular, real problem. This is likely due to discontinuities in the function: in some parts of the space a small perturbation can cause the robot to fall and significantly reduce the reward, while in other parts the reward will be much smoother. Expert tuning of the tripod gait can achieve reward values above 40, much better than any gait found in the optimizations here. ALEBO enables linear embedding methods to reach their full potential, but these results show that there is still much room for additional work in HDBO.

# 8   Conclusion

Our work highlights the importance of two basic requirements for an embedding to be useful for optimization that are often not examined critically by the literature: 1) the function must be well-modeled on the embedding; and 2) the embedding should contain an optimum. To the first point, we showed how polytope constraints on the embedding eliminate boundary distortions, and we derived a Mahalanobis kernel appropriate for GP modeling in a linear embedding. To the second, we developed an approach for computing the probability that the embedding contains an optimum, which we used to construct embeddings with a high chance of containing an optimum, via hypersphere sampling and selecting $d_e > d$. With ALEBO we verified empirically that addressing these issues resolved the poor REMBO performance on benchmark tasks and real-world problems.

These same considerations are important for any embedding. When constructing a VAE for BO it will be equally important to ensure the function remains well-modeled in the embedding, to handle box bounds in an appropriate way, and to ensure the embedding has a high chance of containing an optimum. End-to-end learning of the embedding and the GP can produce an embedding amenable to modeling, though potentially requiring a large number of evaluations to learn the embedding. Adapting nonlinear embeddings learned from auxiliary data for HDBO is an important area of future work. Clipping to box bounds will hurt modelability in a nonlinear embedding in the same way as in a linear embedding. Here we incorporated linear constraints into the acquisition function optimization; for a VAE these constraints will be nonlinear, but their gradients can be backpropped and so constrained optimization can be done in a similar way.

The experiment results show that linear embedding HDBO, and ALEBO in particular, can be a valuable tool for high-dimensional optimization. On real-world problems, we found that local-search methods (CMA-ES and TuRBO) can be highly competitive, or even best-performing. These particular problems have discontinuities that make global modeling difficult and favor local search, however there are other settings where embedding HDBO will be the best choice. The appeal of using embeddings for HDBO is that all of the BO techniques developed over the past two decades can be directly applied to high-dimensional problems. Settings where BO is not matched by local search include cost-aware [44], multi-task [46], and multi-fidelity [52] optimization. These methods can be directly adapted to HDBO by applying them inside a random linear embedding, and the techniques described in this paper will ensure the best possible performance.

## Broader Impact

Bayesian optimization is a powerful optimization technique used in a wide range of industries and applications, such as robotics [32, 6, 40], internet tech companies [16, 30], designing novel molecules for pharmaceutics [17], material design for increasing efficiency of solar cells [54], and aerospace engineering [28]. All of these settings have high-dimensional optimization problems, and advances in BO will reflect on improved capabilities on these fields as well. We have fully open-sourced our code for ALEBO to be available for researchers and practitioners in these fields, and many others. The ability to optimize a larger number of parameters than has previously been possible will bring further improvements to and further accelerate work in these areas.

## Acknowledgements

R.C. thanks Marc Deisenroth and Frank Hutter for insightful discussions, and Victor-Philipp Negoescu, Mark Prediger, Florian Schnell for preliminary experiments back in 2013. B.L. thanks Matthias Feurer for helpful comments.

## Funding Disclosure

We did not receive any third party funding or third party support for this work.

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
