[Supplementary Material]

# Supplemental Materials: Re-Examining Linear Embeddings for High-Dimensional Bayesian Optimization

This supplemental material contains a number of additional results and analyses to support the main text.

## S1  HeSBO Embeddings

We consider HeSBO embeddings in the case of a random axis-aligned true subspace, and a uniform prior on the location of the optimum within that subspace. As explained in Sec. 4, with $d = 2$ and this prior, regardless of $d_e$ or $D$ there are three possible embeddings: (1) each of the active parameters are captured by a parameter in the embedding; (2) the embedding is constrained to the diagonal $x_{i_1} = x_{i_2}$; or (3) the embedding is constrained to the diagonal $x_{i_1} = -x_{i_2}$. Fig. S1 shows these three embeddings for the Branin problem from the top row of Fig. 1.

Within the first embedding, the optimal value of 0.398 can be reached. Within the second, the best value is 0.925 and within the third it is 17.18. Under a uniform prior on the location of the optimum within a random axis-aligned true subspace, it is easy to compute the probability that the HeSBO embedding contains an optimum:

$$P_{\text{opt}}(d_e) = \frac{d_e!}{(d_e - d)! d_e^d}. \tag{S1}$$

For $d = 2$, this is exactly the probability of the first embedding shown in Fig. S1. This probability does not depend on $D$, but increases with $d_e$, and is the probability shown in Fig. 4.

## S2  The Mahalanobis Kernel

When fitting the Mahalanobis kernel derived in Proposition 1, we use an approximate Bayesian treatment of $\boldsymbol{\Gamma}$ to improve model performance while still maintaining tractability. We propagate uncertainty in $\boldsymbol{\Gamma}$ into the GP posterior by first constructing a posterior for $\boldsymbol{\Gamma}$ using a Laplace approximation with a diagonal Hessian, and then drawing $m$ samples from that posterior. The marginal posterior for $f(\boldsymbol{y})$ can then be approximated as:

$$p(f(\boldsymbol{y})) \approx \frac{1}{m} \sum_{i=1}^{m} p(f(\boldsymbol{y})|\boldsymbol{\Gamma}^i).$$

Because of the GP prior, each conditional posterior $p(f(\boldsymbol{y})|\boldsymbol{\Gamma}^i)$ is a normal distribution with known mean $\mu_i$ and variance $\sigma_i^2$. Thus the posterior $p(f(\boldsymbol{y}))$ is a mixture of Gaussians, which we can

Figure S1: Three possible HeSBO embeddings of the $d = 2$ Branin function. *(Left)* The first embedding fully captures the function, and thus captures all three optima. *(Middle)* The second is restricted to the subspace $x_1 = -x_2$. This subspace does not contain an optimum, but comes fairly close. *(Right)* The third embedding is restricted to the subspace $x_1 = x_2$ and does not come close to any optimum.

Figure S2: Test-set model predictions for three GP kernels on the same train/test data generated by evaluating the Hartmann6 $D$=100 function on a fixed linear embedding. A typical ARD kernel fails to learn and predicts the mean. The Mahalanobis kernel predicts well, and posterior sampling is important for getting reasonable predictive variance.

approximate using moment matching:

$$p(f(\boldsymbol{y})) \approx \mathcal{N}\left(\frac{1}{m}\sum_{i=1}^{m}\mu_i, \frac{1}{m}\sum_{i=1}^{m}\sigma_i^2 + \mathrm{Var}_i[\mu_i]\right).$$

We do this to maintain a Gaussian posterior, under which acquisition functions like EI have analytic form and can easily be optimized, even subject to constraints as in (1).

As described in Sec. 5, we show the importance of the Mahalanobis kernel using models fit to data from the Hartmann6 $D$=100 function. We generated a projection matrix $\boldsymbol{B}$ using hypersphere sampling to define a 6-d linear embedding. We then generated a training set (100 points) and a test set (50 points) within that embedding—that is, within the polytope given by (1)—using rejection sampling. We fit three GP models with different kernels to the training set, and then evaluated each on the test set: a typical ARD RBF kernel in 6 dimensions, the Mahalanobis kernel using a point estimate for $\boldsymbol{\Gamma}$, and the Mahalanobis kernel with posterior marginalization for $\boldsymbol{\Gamma}$ as described above.

Fig. S2 compares model predictions for each of these models with the actual test-set outcomes; results here are the same as in Fig. 3 with the addition of the Mahalanobis point estimate kernel. With an ARD RBF kernel, the GP predicts the function mean everywhere, which is typical behavior of a GP that has failed to learn the function. With the same training data, the Mahalanobis kernel is able to make accurate predictions on the test set. Using a point estimate for $\boldsymbol{\Gamma}$ significantly underestimates the predictive variance, which is rectified by using posterior sampling as described above. In BO exploration is driven by model uncertainty, so well-calibrated uncertainty intervals are especially important.

Fig. S3 evaluates the predictive log marginal probabilities for the ARD RBF kernel and the Mahalanobis kernel with posterior sampling across a wide range of training sets with different sizes (without posterior sampling, Fig. S2 shows that the Mahalanobis point estimate significantly under covers and so has very poor predictive log marginal probabilities). We used the same linear embedding and Hartmann6 $D$=100 function used in Fig. S2 to sample 1000 test points which were held fixed. For each of 8 training set sizes ranging from 40 to 200, we randomly sampled 20 training sets from the embedding. For each training set, we fit the two GPs, made predictions on the 1000 test points, and then computed the average marginal log probability of the true values. Fig. S3 shows that as the training set size increased from 40 to 200, the ARD RBF kernel could only improve slightly on predicting the mean, as it did in Fig. S2; even 200 points in the 6-d embedding were not sufficient to significantly improve the model. For small training set sizes, the Mahalanobis kernel (with sampling) had high variance in log likelihood, as it has the potential to overfit and thus under cover. But for training set sizes of 50 and greater it had better predictive log likelihood than the ARD RBF kernel, and continued to learn as the training set size was increased. For small datasets, the Mahalanobis kernel can overfit and thus have poor predictive likelihood, but for the purposes of BO, overfitting can be better than not fitting at all (predicting the mean), even when predicting the mean has better

Figure S3: Average test-set log likelihood as a function of training set size, for training sets randomly sampled from a fixed linear embedding. Log marginal probabilities were averaged over a fixed test set of 1000 random points. For each training set size, 20 random training sets were drawn of that size and the figure shows the average result over those draws (with error bars for two standard errors). The ARD RBF kernel continues to predict the mean as the training set size is increased, while the Mahalanobis kernel is able to learn as the training set is expanded.

predictive log likelihood. This can be seen in the optimization results (Figs. 5 and S7) where ALEBO showed strong performance even with less than 50 iterations.

## S3   Stationarity in the Embedding

A stationary kernel is one that depends only on $\boldsymbol{x} - \boldsymbol{x}'$, not on the individual values of $\boldsymbol{x}$ and $\boldsymbol{x}'$, and is thus invariant to translation [41]. The following result shows that with linear embeddings, stationarity in the true function implies stationarity in the embedding.

**Proposition S1.** *Suppose the function on the true subspace is drawn from a GP with a stationary kernel:* $f_d \sim \mathcal{GP}(m(\cdot), k(\cdot, \cdot))$ *where* $k(\boldsymbol{z}, \boldsymbol{z}') = \kappa(\boldsymbol{z} - \boldsymbol{z}')$. *Let* $g(\boldsymbol{y}) = \boldsymbol{T}\boldsymbol{B}^\dagger \boldsymbol{y}$. *For any pair of points in the embedding* $\boldsymbol{y}$ *and* $\boldsymbol{y}'$,

$$Cov[f_B(\boldsymbol{y}), f_B(\boldsymbol{y}')] = \tilde{\kappa}(\boldsymbol{y} - \boldsymbol{y}'),$$

*where* $\tilde{\kappa} = \kappa \circ g$. *The implied kernel on the embedding is thus stationary.*

*Proof.* The argument follows that of Prop. 1. As shown there, $f_B(\boldsymbol{y}) = f_d(\boldsymbol{T}\boldsymbol{B}^\dagger \boldsymbol{y})$. Then,

$$\begin{aligned}
\text{Cov}[f_B(\boldsymbol{y}), f_B(\boldsymbol{y}')] &= \text{Cov}[f_d(\boldsymbol{T}\boldsymbol{B}^\dagger \boldsymbol{y}), f_d(\boldsymbol{T}\boldsymbol{B}^\dagger \boldsymbol{y}')] \\
&= \kappa(\boldsymbol{T}\boldsymbol{B}^\dagger \boldsymbol{y} - \boldsymbol{T}\boldsymbol{B}^\dagger \boldsymbol{y}') \\
&= \tilde{\kappa}(\boldsymbol{y} - \boldsymbol{y}').
\end{aligned}$$

$\square$

Consider now clipping to box bounds in the ambient space with the $L^2$ projection $p_{\mathcal{B}}$. Then, $f_B(\boldsymbol{y}) = f_d(\boldsymbol{T}p_{\mathcal{B}}(\boldsymbol{B}^\dagger \boldsymbol{y}))$, and the implied kernel in the embedding is

$$\text{Cov}[f_B(\boldsymbol{y}), f_B(\boldsymbol{y}')] = \kappa(\boldsymbol{T}(p_{\mathcal{B}}(\boldsymbol{B}^\dagger \boldsymbol{y}) - p_{\mathcal{B}}(\boldsymbol{B}^\dagger \boldsymbol{y}'))).$$

This is clearly non-stationary, because $p_{\mathcal{B}}$ is not translation invariant.

## S4   Polytope Bounds on the Embedding

Rather than using projections to the box bounds $\mathcal{B}$, we specify polytope constraints in (1). Fig. S4 illustrates the embedding with these constraints for the same Branin $D = 100$ problem from the top row of Fig. 1. The embedding in the left figure was created with the REMBO strategy of sampling each entry from $\mathcal{N}(0, 1)$. For the embedding in the right figure, that same projection matrix had

Figure S4: *(Left)* An embedding from a $\mathcal{N}(0,1)$ projection matrix on the same Branin $D = 100$ problem from Fig. 1 subject to constraints of (1). *(Right)* The embedding from the same projection matrix after normalizing the columns to produce unit circle samples. Sampling from the unit circle increases the probability that an optimum will fall within the embedding, and polytope bounds avoid nonlinear distortions.

each column normalized. This converts the projection matrix to be a sample from the unit circle, as described in Sec. 4.

The $\mathcal{N}(0,1)$ embedding does not contain any optima within the polytope bounds. Converting that projection matrix to a hypersphere sample rounds out the vertices of the polytope and expands the space to capture two of the optima. Consistent with Fig. 4, we see that hypersphere sampling significantly improves the chances of the embedding containing an optimum. Fig. S4 also shows that with the polytope bounds, we avoid the nonlinear distortions seen with REMBO in Fig. 1.

Note that adding linear constraints to a non-convex optimization problem (acquisition function optimization) does not change the complexity of that problem.

## S5 Evaluating the Probability the Embedding Contains an Optimum

As in other parts of the paper, we consider a uniform prior on the location of the optimum within a random axis-aligned subspace. A random true projection matrix $\boldsymbol{T}$ is sampled by selecting $d$ columns at random and setting each to one of the $d$-dimensional unit vectors. $\boldsymbol{z}^*$ is then sampled uniformly at random from $[-1, 1]^d$. $\boldsymbol{B}$ is sampled according to the desired strategy, which in our experiments was REMBO ($\mathcal{N}(0,1)$ entries), HeSBO, or hypersphere. Given these three quantities, we can evaluate whether or not the embedding contains an optimum that satisfies the constraints of (1) by solving the following linear program:

$$\begin{aligned} \text{maximize } & \boldsymbol{0}^\top \boldsymbol{x} \\ \text{subject to } & \boldsymbol{T}\boldsymbol{x} = \boldsymbol{z}^*, \\ & (\boldsymbol{B}^\dagger \boldsymbol{B} - \boldsymbol{I})\boldsymbol{x} = \boldsymbol{0}, \\ & \boldsymbol{x} \geq -\boldsymbol{1}, \\ & \boldsymbol{x} \leq \boldsymbol{1}. \end{aligned}$$

If this problem is feasible, then the embedding produced by $\boldsymbol{B}$ contains an optimum; if it is infeasible, then it does not. Solving this over many draws of $\boldsymbol{T}$, $\boldsymbol{z}^*$, and $\boldsymbol{B}$ produces an estimate of $P_{\text{opt}}$ under that prior for the location of optima. Here we used a uniform prior, but this linear program can be taken to compute $P_{\text{opt}}$ under any prior.

Fig. S5 shows $P_{\text{opt}}$ for the three embedding strategies as a function of $d$ and $d_e$, for $D$ fixed at 100. The results shown for $d = 2$ and $d = 6$ are those given in the main text in Fig. 4. Fig. S6 shows $P_{\text{opt}}$ for a wide range of values of $d$ and $D$, for hypersphere sampling. Across this wide range we see that for many values of $d$ we can achieve high values of $P_{\text{opt}}$ with reasonable values of $d_e$, even for relatively high values of $D$.

Figure S5: $P_{\text{opt}}$ as estimated in Fig. 4, extended with the results for $d = 10$. Setting $d_e > d$ significantly improves the probability of the embedding containing an optimum.

Figure S6: $P_{\text{opt}}$ for hypersphere sampling, as estimated in Fig. 4 but here for a wider range of values of $d$ and $D$. Contour color indicates $P_{\text{opt}}$. Doubling $D$ decreases $P_{\text{opt}}$ for $d$ and $d_e$ fixed, however even at $D = 200$, high values of $P_{\text{opt}}$ with reasonable values of $d_e$ can be had for many values of $d$.

## S6 Selecting the Embedding Dimension

Linear embedding HDBO requires selecting a dimensionality for the embedding. The results in Sections 5 and S5 show clearly that choosing an embedding dimensionality higher than that of the true subspace is vital for obtaining a high probability of the the embedding containing an optimum. In principal, if one knew the true subspace dimension $d$, the results of Sec. S5 could be used to calculate $P_{\text{opt}}$ as a function of $d_e$, and then $d_e$ could be chosen to reach a desired value of $P_{\text{opt}}$. In practice, however we will not typically know what the true subspace dimension is, or even be certain of the existence of a true, linear subspace.

In real BO problems, which have expensive function evaluations, there is always a sample budget that depends on the function evaluation cost and available resources. A simulation that takes several minutes may allow a few hundred iterations, as in the Daisy experiment of Sec. 7. When function evaluations are A/B tests that take around a week, the evaluation budget may be limited to less than 50 iterations [*e.g.*, 29]. Generally in BO, there is a trade-off between the number of parameters one can optimize and the number of iterations that will be required, and in real problems one must select the number of parameters according to the evaluation budget. In that sense, there is no difference with linear embedding HDBO: $d_e$ should be set to the highest value that is supported by the available evaluation budget. With a budget of 50 iterations and $d_e = 15$, it will be unlikely to get good model fit quickly enough to effectively optimize, so smaller values like 8 or 10 would be warranted. On the other hand, with the 500 iteration budget of the Daisy problem, one could set $d_e$ in the 15–20 range (the maximum supported by normal BO) to maximize $P_{\text{opt}}$.

Simulations and model cross-validation can be helpful for identifying the maximum number of parameters that can be effectively tuned for a particular evaluation budget, but there has been little work in this area. The nature of the dimensionality vs. iteration budget trade-off is important in all real BO problems, not just with linear embedding HDBO, so appropriate heuristics for this question is an important area of future work.

## S7 Handling Black-Box Constraints in High-Dimensional Bayesian Optimization

In many applications of BO, in addition to the black-box objective $f$ there are black-box constraints $c_j$ and we seek to solve the optimization problem

$$\text{minimize } f(\boldsymbol{x})$$
$$\text{subject to } c_j(\boldsymbol{x}) \leq 0, \quad j = 1, \ldots, J,$$
$$\boldsymbol{x} \in \mathcal{B}.$$

In most settings the constraint functions $c_j$ are evaluated simultaneously with the objective $f$. Constraints are typically handled in BO by fitting a separate GP to each outcome (that is, to $f$ and to each $c_j$). The acquisition function is then modified to consider not only the objective value but also whether the constraints are likely to be satisfied [*e.g.*, 14].

The extension of BO in an embedding to constrained BO is straightforward, so long as the same embedding is used for every outcome. A separate GP (in the case of ALEBO, using the Mahalanobis kernel) is fit to data from each outcome. Because the embedding is shared, predictions can be made for all of the outcomes at any point in the embedding. This allows us to evaluate and optimize an acquisition function for constrained BO in the embedding. Once a point is selected, it is projected up to the ambient space and evaluated on $f$ and each $c_j$ as usual. Random projections are especially well-suited for constrained BO because there is no harm in requiring the same projection for all outcomes, since it is a random projection anyway.

These same considerations apply to multi-objective optimization. Acquisition functions for multi-objective optimization can be directly applied to HDBO using linear embeddings in the same way that those for constrained optimization are used here.

## S8 Additional Benchmark Experiment Results

Here we provide results from two additional benchmark problem (Hartmann6 $D$=100, and Hartmann6 random subspace $D$=1000), three additional methods (LineBO variants), and provide a study of the sensitivity of ALEBO performance to $d_e$ and $D$. We also provide implementation details for the experiments, and an extended discussion of the results from each experiment.

### S8.1 Method Implementations and Experiment Setup

The linear embedding methods (REMBO, HeSBO, and ALEBO) were all implemented using BoTorch, a framework for BO in PyTorch [2], and so used the same acquisition functions and the same tooling for optimizing the acquisition function. Importantly, this means that all of the difference seen between the methods in the empirical results comes exclusively from the different models and embeddings. EI was the acquisition function for the Hartmann6 and Branin benchmarks, and NEI [30] was used to handle the constraints in the Gramacy problem. ALEBO and HeSBO were given a random initialization of 10 points, and REMBO was given a random initialization of 2 points for each of its 4 projections used within a run.

The remaining methods used reference implementations from their authors with default settings for the package: REMBO-$\phi k_\Psi$ and REMBO-$\gamma k_\Psi$[1]; EBO[2]; Add-GP-UCB [3]; SMAC[4]; CMA-ES[5]; and

CoordinateLineBO, RandomLineBO, and DescentLineBO[6]. EBO requires an estimate of the best function value, and for each problem was given the true best function value. SMAC and CMA-ES require an initial point, and were given the point at the center of the ambient space box bounds. See the benchmark reproduction code at `github.com/facebookresearch/alebo` for the exact calls used for each method.

The function evaluations for all problems were noiseless, so the stochasticity throughout the run and in the final value all comes from stochasticity in the methods themselves. For linear embedding methods the main sources of stochasticity are in generating the random projection matrix and in the random initialization.

### S8.2 Analysis of experimental results

Fig. S7 provides a different view of the benchmark results of Fig. 5, showing log regret for each method, averaged over runs with error bars indicating two standard errors of the mean. This is evaluated by taking the value of the best point found so far, subtracting from that the optimal value for the problem, and then taking the log of that difference. The results are consistent with those seen in Fig. 5, and the standard errors show that ALEBO's improvement in average performance over the other methods is statistically significant. We now discuss some specific aspects of these experimental results.

**Branin $D$=100** Starting from around iteration 20, ALEBO performed the best of all of the methods. The distribution of final iteration values shows that in one iteration the ALEBO embedding did not contain an optimum and so achieved a final value near 10. However, across all 50 runs nearly all achieved a value very close to the optimum, leading to the best average performance. Without the log transform (Fig. 5), SMAC and the additive GP methods were the next best performing.

The poor performance of HeSBO on this problem (particularly in Fig. 5 without the log, where it is outperformed by all methods other than Sobol) can be attributed entirely to the embedding not containing an optimum. Recall that for this problem there are exactly three possible HeSBO embeddings, which are shown in Fig. S1. As explained in Sec. S1, the first embedding contains the optimum of 0.398, while the best value in the other embeddings are 0.925 and 17.18. Thus, if the BO were able to find the true optimum within each embedding with the budget of 50 function evaluations given in this experiment, the expected best value found by HeSBO would be:

$$0.398 P_{\text{opt}} + 0.925 \left( \frac{1 - P_{\text{opt}}}{2} \right) + 17.18 \left( \frac{1 - P_{\text{opt}}}{2} \right).$$

This is the best average performance one can hope to achieve using the HeSBO embedding on this problem. Using (S1) we can compute $P_{\text{opt}}$ for $d_e = 4$ as 0.75, and it follows that the HeSBO expected best value is 2.56. This is nearly exactly the average best-value shown in Fig. 5. The poor performance of HeSBO is thus not related to BO, but comes entirely from the 12.5% chance of generating an embedding whose optimal value is 17.18. The presence of these embeddings can be clearly seen in the distribution of final best values in Fig. 5.

**Hartmann6 $D$=1000** As noted in the main text, the additive kernel methods and REMBO-$\gamma k_\Psi$ could not scale up to the 1000 dimensional problem. SMAC also became very slow and was only run for 10 repeats (rather than 50) on the $D$=1000 problems. A nice property of linear embedding approaches is that the running time is not significantly impacted by the ambient dimensionality. Table S1 gives the average running time per iteration for the various benchmark methods (all run on the same 1.90GHz processor and allocated a single thread). Inferring the additional parameters in the Mahalanobis kernel and the added linear constraints make ALEBO slower than other linear embedding methods, but it is faster than the additive kernel methods (an order of magnitude faster than Add-GP-UCB), and at $D$=1000 is an order of magnitude faster than SMAC. The average of about 50s per iteration is short relative to the function evaluation time of typical resource-intensive BO applications.

On both this problem and the $D$=100 version, REMBO performed worse than Sobol, despite there being a true linear subspace that satisfies the REMBO assumptions. The source of the poor perfor-

Figure S7: Log regret for the benchmark experiments of Fig. 5, plus Hartmann6 with $D$=100 and with a random (non-axis-aligned) subspace in $D$=1000. Each trace is the mean over 50 repeated runs, with errors bars showing two standard errors of the mean. ALEBO was a best-performing method on all problems; on Hartmann6 $D$=100 it tied with REMBO-$\gamma k_\Psi$, TuRBO, and SMAC as the best methods.

Table S1: Average running time per iteration in seconds on the Hartmann6 problem, $D$=100 and $D$=1000.

| | $D$=100 | $D$=1000 |
|---|---|---|
| ALEBO | 42.7 | 52.5 |
| REMBO | 1.6 | 1.9 |
| HeSBO, $d_e$=$d$ | 1.1 | 2.0 |
| HeSBO, $d_e$=2$d$ | 1.1 | 2.3 |
| REMBO-$\phi k_\Psi$ | 2.1 | 1.1 |
| REMBO-$\gamma k_\Psi$ | 7.2 | — |
| EBO | 69.6 | — |
| Add-GP-UCB | 995.0 | — |
| SMAC | 26.2 | 1137.9 |
| CMA-ES | 0.0 | 0.1 |
| Sobol | 0.1 | 0.8 |

mance is the poor representation of the function on the embedding illustrated in Fig. 1. Correcting these issues as is done in ALEBO significantly improves the performance.

**Hartmann6 $D$=100** ALEBO, REMBO-$\gamma k_\Psi$, TuRBO, and SMAC were the best-performing methods on this problem. HeSBO and Add-GP-UCB both did very well early on, but then got stuck and did not progress significantly after about iteration 50. For HeSBO, this is likely because the performance is ultimately limited by the low probability of the embedding containing an optimum.

This problem was used to test three additional methods beyond those in Fig. 5: CoordinateLineBO, RandomLineBO, and DescentLineBO [27]. These are recent methods developed for high-dimensional safe BO, in which one must optimize subject to safety constraints that certain bounds on the functions must not be violated. The performance of these methods can be seen in the fourth panel of Fig. S7: all three LineBO variants perform much worse than Sobol, and show almost no reduction of log regret. This finding is consistent with the results of Kirschner et al. [27], who used the Hartmann6 $D$=20 problem as a benchmark problem. At $D$=20, they found that CoordinateLineBO required about 400 iterations to outperform random search, and even after 1200 iterations RandomLineBO and DescentLineBO did not perform better than random search. These methods are designed specifically for safe BO, which is a significantly harder problem than usual BO that has much worse scaling with dimensionality. The primary challenge for high-dimensional safe BO lies in optimizing the acquisition function, which is difficult even for relatively small numbers of parameters where there is no difficulty in optimizing the traditional BO acquisition function. The LineBO methods develop new techniques for acquisition function optimization, but do not consider difficulties with GP modeling in high dimensions, which is the main focus of HDBO work. LineBO methods perform very well on safe BO problems relative to other methods, but ultimately non-safe HDBO is not the problem that they were developed for, and so it is not surprising to see that they were not successful on this task.

**Hartmann6 random subspace $D$=1000** Linear embedding BO methods assume the existence of a true linear subspace, but do not assume anything about the orientation of that subspace and are generally invariant to rotation. Prior work on HDBO has typically focused on the axis-aligned (unused variables) problems that we used here, but we also include a non-axis-aligned problem. We generated a random true embedding by sampling a rotation matrix from the Haar distribution on the special orthogonal group $\mathrm{SO}(D)$ [45], and then taking the first $d$ rows to specify a projection matrix $\boldsymbol{T}$ from $D = 1000$ down to $d = 6$. This defines a non-axis-aligned true subspace, and we took the true low-dimensional function $f_d$ as the Hartmann6 function on this subspace. Bayesian optimization proceeded as with the other problems, and results for the linear subspace methods were similar to the axis-aligned $D$=1000 problem, except REMBO performed equal to HeSBO.

### S8.3 Sensitivity of ALEBO to Embedding and Ambient Dimensions

We study sensitivity of ALEBO optimization performance to the embedding dimension $d_e$ and the ambient dimension $D$ using the Branin function. To test dependence on $d_e$, for $D = 100$ we ran 50 optimization runs for each of $d_e \in \{2, 3, 4, 5, 6, 7, 8\}$. To test dependence on $D$, for $d_e = 4$ we ran

Figure S8: ALEBO performance on the Branin problem, (*Left*) as a function of embedding dimension $d_e$ and (*Right*) as a function of ambient dimension $D$. Performance shown is the average of 50 repeated runs. Optimization performance is poor with $d_e = 2$, but shows little sensitivity to $d_e$ for values greater than 2. Optimization performance shows little sensitivity with $D$, all the way up to $D = 1000$.

Figure S9: Final best value for the Branin problem optimizations of Fig. S8, as mean with error bars showing two standard errors. With the exception of $d_e$ of 2 or 3, optimization performance was good across a wide range of values of $d_e$ and $D$.

50 optimization runs for each of $D \in \{50, 100, 200, 500, 1000\}$. Note that the $d_e = 4$ and $D = 100$ case in each of these is exactly the optimization problem of Fig. 5.

The results of the optimizations are shown in Figs. S8 and S9. For $d_e = d$, optimization performance was poor. From Fig. 4 we know this is because there is a low probability of the embedding containing an optimizer. Increasing $d_e$ increases that probability, but also increases the dimensionality of the embedding and thus reduces the sample efficiency of the BO in the embedding. This trade-off can be seen clearly in Fig. S8: with $d_e = 2$ there is rapid improvement that then flattens out because of the lack of good solutions in the embedding, whereas for $d_e = 8$ the initial iterations are worse but then it ultimately is able to find much better solutions. Even at $d_e = 8$ the average best final value was better than that of any of the comparison methods in Fig. 5.

The ambient dimension $D$ will not directly impact the GP modeling in ALEBO, which depends only on $d_e$, however it will impact the probability the embedding contains an optimum as shown in Fig. S6. Consistent with the strong ALEBO performance for the Hartmann6 $D=1000$ problem, we see here that even increasing $D$ to 1000 does not significantly alter optimization performance. Even at $D = 1000$, ALEBO had better performance than the other benchmark methods had on $D = 100$.

Figure S10: Ablation study results comparing full ALEBO (same trace from Fig. 5) with: (1) Ablation of the Mahalanobis kernel, replacing it with an ARD Matern 5/2 kernel, and (2) Ablation of the hypersphere-sampled projection matrix, replacing it with an i.i.d. normal random matrix. Both components are necessary for the good HDBO performance of ALEBO, with the Mahalanobis kernel the most important factor.

### S8.4 Ablation Study

We use an ablation study to better understand the impact of two of the new developments incorporated into ALEBO: the Mahalanobis kernel for improved modeling in the embedding, and the hypersphere sampling for increasing the probability that the embedding contains an optimum. We used the Branin $D = 100$ problem for this study, with $d_e = 4$ as in Fig. 5. For the ablation of the kernel, we replaced the Mahalanobis kernel with an ARD Matern 5/2 kernel. For the ablation of the sampling, we replaced hypersphere sampling with the random normal samples used by REMBO. Results are given in Fig. S10. Removing either component significantly decreased BO performance, and removing the Mahalanobis kernel was especially detrimental.

## S9 Constrained Neural Architecture Search Problem

NAS-Bench-101[53] is a dataset of convolutional neural network (CNN) performance on the CIFAR-10 problem, produced for the purpose of reproducible research in neural architecture search. The search space is to design the cell for a CNN using a DAG with 7 nodes and up to 9 edges. The first node is input and the last node is output; the remaining five can be selected to be any one of the operations $3 \times 3$ convolution, $1 \times 1$ convolution, or $3 \times 3$ max-pool. The edges connect these operations to each other, and to the input and output. This space includes more than 400,000 unique models, each of which was evaluated on CIFAR-10 by training for 108 epochs on a TPU, and then testing on the test set. For each model, several metrics were computed, including the number of seconds required for training and the final test-set accuracy.

We parameterized this as a continuous HDBO problem by separately parameterizing the operations and edges. The operations were parameterized using one-hot encoding, which, with five selectable nodes and 3 options for each, produced 15 parameters. These were optimized in the continuous $[0, 1]$ space, and then the max for each set was taken as the "hot" feature that specified which operation to use in the corresponding node. NAS-Bench-101 represents the edges using the upper-triangular $7 \times 7$ adjacency matrix, which has $\frac{(7 \cdot 6)}{2} = 21$ binary entries. These entries were similarly optimized in the continuous $[0, 1]$ space, and the adjacency matrix was created iteratively by adding entries in the rank order of their corresponding continuous-valued parameters, and finding the largest number of non-zero entries that can be added to still have no more than 9 edges after pruning portions not connected to input or output. The combination of the adjacency matrix parameters (21) and the one-hot-encoded operation parameters (15) produced a 36-dimensional optimization space.

Figure S11: Optimization performance on the hexapod locomotion task from Fig. 7. *(Left)* Each trace shows the best value by each iteration, averaged across repeated runs with error bars showing two standard errors. *(Right)* The distribution of values at the final iteration. ALEBO performed best of the HDBO methods, but CMA-ES outperformed them all.

The objective for the optimization was to maximize test-set accuracy, subject to a constraint on training time being less than 30 minutes. A large portion of the models in the NAS-Bench-101 modeling space have training times above 30 minutes, with the longest around 90 minutes. While test-set accuracy is proportional to training time [53], our results in Fig. 6 show that with HDBO it is possible to find well-performing models with short training times. TuRBO and CMA-ES were adapted to this constrained problem by return a poor objective value for infeasible points.

In the results of Fig. 6, HeSBO showed strong performance in early iterations, but quickly flattened out and ended up worse than random on average. By the final iteration, ALEBO performed significantly better than Sobol, CMA-ES, HeSBO, and REMBO. TuRBO performed worse on early iterations, but in later iterations had performance that was not significantly different from ALEBO.

## S10   Locomotion Optimization Problem

The task for the final set of experiments was to learn a gait policy for a simulated robot. As a controller, we use the Central Pattern Generator (CPG) from [9]. The goal in this task is for the robot to walk to a target location in a given amount of time, while reducing joint velocities, and average deviation from a desired height

$$f(\boldsymbol{p}) = C - ||\boldsymbol{x}_{\text{final}} - \boldsymbol{x}_{\text{goal}}|| - \sum_{t=0}^{T} \left( w_1 ||\dot{\boldsymbol{q}}_t|| - w_2 |h_{\text{robot},t} - h_{\text{target}}| \right), \tag{S2}$$

where $C = 10$, $w_1 = 0.005$, and $w_2 = 0.01$ are constants. $\boldsymbol{x}_{\text{final}}$ is the location of the robot on a plane at the end of the episode, $\boldsymbol{x}_{\text{goal}}$ is the target location, $\dot{\boldsymbol{q}}_t$ are the joint velocities at time $t$ during the trajectory, $h_{\text{robot},t}$ is the height of the robot at time $t$, and $h_{\text{target}}$ is a target height. $T = 3000$ is the total length of the trajectory, leading to $30s$ of experiment. The objective function is evaluated at the end of the trajectory.

Fig. S11 shows the optimization performance over 50 repeated runs, which are the same results of Fig. 7 but including errors bars and the distribution of final best values. All of the methods have high variance in their final best value across runs. ALEBO has the lowest variance and thus the most robust performance. SMAC was able to find a good value in one run, but on average performed slightly worse than ALEBO. TuRBO performed better, but CMA-ES was the clear best-performing method on this problem.

## Footnotes

[1] `github.com/mbinois/RRembo`

[2] `github.com/zi-w/Ensemble-Bayesian-Optimization`

[3] `github.com/dragonfly/dragonfly`, with option `acq="add_ucb"`

[4] `github.com/automl/SMAC3`, SMAC4HPO mode

[5] `github.com/CMA-ES/pycma`

[6]`github.com/jkirschner42/LineBO`