[Reviews · NeurIPS 2020]

Review 1

Summary and Contributions: High-dimensional optimization problems are still challenging for Bayesian optimization methods. Linear embeddings are one alternative to handle this setting, but suffer from several limitations: (i) the embedding might produce representations that are hard to model with a Gaussian Process, (ii) points chosen in the embedding map outside the bounds of the optimization problem, (iii) product kernels are not preserved in the embedded space and (iv) the embedding might not contain the optimum. This paper studies these limitations and proposes a solution, called ALEBO, (i) using a different kernel, (ii) searching only that portion of the embedded space that can be mapped to the original bounds and (iii) maintaining a high probability that the optimum lies in the embedding. The paper compares ALEBO to existing high-dimensional BO methods, such as REMBO, SMAC, TurBO and HeSBO but also other global optimization methods such as CMA-ES.

Strengths: High-dimensional optimization with a high sample efficiency, e.g. for tuning hyperparameters of machine learning models, is still challenging and a highly relevant topic for the NeurIPS community. REMBO and HeSBO make use of embeddings and thus approach high-dimensional problems. However, these methods still suffer some limitations. This paper explains and studies what causes these limitations, how they affect practical applications and propose a solution that overcomes these problems.

Weaknesses: The paper proposed a convincing method, but I have two criticisms: While the paper proposed a thorough analysis and gives detailed explanations of limitations of prior work, the overall contribution is rather incremental and does not yield a highly competitive baseline compared to TurBO and CMA-ES: On the constrained NAS problem not all methods were evaluated (see my comments below) and on the policy search benchmark another BO method, TuRBO, seems to perform better. Furthermore, there also remains the open question of how to select the dimensionality of the embedding in practice, which potentially dramatically impacts the optimization performance.

Correctness: The claims and method seem correct to me.

Clarity: The paper is overall well written, however, parts of the supplemental are critical to understand the paper. I am aware of the space restrictions, but I would recommend to move Algorithm 1 to the main paper since I found it very helpful to understand how the different parts of the method work together. Furthermore, I would also replace Figure 7 (right) with Figure S11 from the supplemental since it shows the variance across repetitions.

Relation to Prior Work: To the best of my knowledge prior work is properly addressed and discussed.

Reproducibility: Yes

Additional Feedback: Q1: Does the NASBench 101 benchmark require using the constraint that the training time is less than 30min? I agree, that constrained optimization is an important topic in BO, but in my opinion, this is not the main focus of this paper. This design decision does not allow to compare to other optimizers and so I am wondering how these would compare on this benchmark if there are no constraints? Q2: How would one set d_e in practice. Do you have any insights whether ALEBO is more or less sensitive wrt. to this hyperparameter compared to prior work? ## After readings the author's response I'd like to thank the authors for addressing my questions and concerns. I appreciate adding a discussion about the flexibility of the proposed method and extending the set of optimization methods compared on the the real-world problems, which, I believe, will be a great addition and significantly strengthens the contribution.


Review 2

Summary and Contributions: In this article the authors revisit the use of random linear embeddings in Bayesian optimization. They identify key aspects explaining some mitigated results in benchmarks. Improvements include restriction to polytope bounds and the use of an appropriate kernel to model projected observations (and handle distortion). Benchmark results on synthetic and realistic problems show promising results.

Strengths: I found the paper to be well structured and pedagogical given the page limit. The performance of the proposed method is good over several tests and include relevant competitors.

Weaknesses: It could be worth detailing more the limitations: arbitrary selection of de, longer training time with Mahalanobis kernel, limits of linear embeddings.

Correctness: The paper comes with backing simulation results and extensive tests.

Clarity: I would move the algorithmic summary in the paper to fix some ideas (e.g., that B is fixed after sampling once).

Relation to Prior Work: A good summary of existing related works is provided.

Reproducibility: Yes

Additional Feedback: Some statements could be clarified: L95: is k=4 always used? L213: it does for axis-aligned embeddings And I have several questions and suggestions for improvements: a) How to choose de in practice, given that the number of hyperparameters grows quadratically? In the example around line 230, retaining 12 to 20 variables is necessary with a relatively optimistic prior on the location of the optimum (it is not uncommon that several influential linear combinations of variables reach their bounds), but this is already quite high-dimensional for GP modeling. c) For testing the proposed Mahalanobis kernel, rather than testing on the Hartmann 6 function were most variables are inactive, it would be more pertinent to work on a sample from a D-dimensional ARD kernel. Tests could include rotation with just d-active ARD variables (with or without rotation) but also a non-zero activity for the D-d ones to be more realistic. ### Additions after rebuttal ### I thank the authors for addressing my questions. I am also appreciate the propositions for the additional page with more results and discussion. Based on the analysis of pitfalls of random embeddings, this could help defining a better baseline for new high-dim BO methods compared to the naive implementation of REMBO.


Review 3

Summary and Contributions: This paper closely looks into linear embeddings to scale up Bayesian optimization (BO) to high dimensions. The authors give a detailed analysis of why existing linear embedding techniques may fall short, and builds on these insights to propose ALEBO. This new linear embedding approach relies on a Mahalanobis kernel in tandem with a hypersphere sampling to increase the probability that the optimal solution exists in the embedding. Experiments on synthetic data as well as on two real-world scenarios show that ALEBO effectively fixes many of the shortcomings of linear embeddings and improves their performance.

Strengths: 1. This is the first paper to perform an in-depth analysis of several shortcomings of linear embeddings through an extensive empirical evaluation and ablation studies. I expect these insights to benefit the AutoML community and steer future work on high-dimensional BO in the right directions (potentially away from linear embeddings). 2. The paper introduces ALEBO, which alleviates the limitations of linear embedding approaches and is a step forward making them competitive with the state of the art.

Weaknesses: 1. Despite the proposed improvements, the real-world experiments show that linear embeddings still do not emerge as a competitive approach in practice. To some extent, this narrows down the scope and impact of the contribution.

Correctness: The claims are correct and all empirically grounded by ablation studies and comparisons with baselines.

Clarity: The presentation is excellent: well organized, crisp and clear. The paper was a pleasure to read.

Relation to Prior Work: Prior work is carefully discussed and an extensive set of baselines compared against. The experiments do miss some baselines from the family of methods making use of surrogate models to scale up to high dimensions. An example is Rolland et al., 2018, which would have been a strictly better and more interesting baseline compared to Add-GP-UBC.

Reproducibility: Yes

Additional Feedback: I think this is a good paper that will inform future work on high-dimensional BO. Having highlighted a number of severe shortcomings of linear embeddings, I expect future work to either leverage the insights of ALEBO to develop a truly competitive baseline, or simply use these lessons learned to focus on different methods, such as the model-free ones. The robot locomotion experiment does suggest that linear embeddings, despite all improvements, are still not suited to be the default for high dimensional BO. Not only are they outperformed by model-free methods, such as CMA-ES, but also by some model-based ones such as TuRBO (despite the larger variance, as shown in the appendix). In any case, while we do not have a new state of the art method for high-dimensional BO out of this paper, the contribution is useful and will inform future work in this space. Some questions are below. 1. Figure 6: only a subset of methods are benchmarked here compared to Figure 7. I assume the reason is that only methods that support constrained optimization are considered. Have the authors also tried real-world experiments where the stronger baselines, such as CMA-ES, can actually be applied? How does ALEBO compare to them in these settings? Is it still outperformed as in Figure 7? 2. As Figure 6 focuses on a constrained optimization problem, how would approaches based on simple BO with constrained acquisition functions perform? Some examples include constrained expected improvement, constrained predictive entropy search, and constrained max-value entropy search (Gardner et al., 2014; Lobato et al., 2015; Perrone et al., 2019). 3. The authors focused the discussion on linear embedding methods, leaving nonlinear methods out of the scope of the work. Still, I would have liked to see more discussion on nonlinear methods based on VAEs. How do these compare to ALEBO? What learnings about the shortcomings of linear embeddings transfer to the nonlinear case? 4. Table S1 in the appendix shows that ALEBO's runtime scales very well in the problem dimension, but otherwise does not compare particularly well with several baselines. I suppose this is due to the complications implied by the linear constraints and the Mahalanobis kernel, which requires MCMC sampling and moment matching. How would ALEBO perform when the x axis is wall-clock time rather than iteration count? CMA-ES and Sobol would be easily winners when blackbox evaluations incur reasonable runtime, but then also REMBO and HeSBO could be more competitive considering they can take up to 40 times less compute time per iteration. *** Post-rebuttal *** I'd like to thank the authors for addressing my questions. I also appreciate that there will be an additional discussion around the connection with non-linear embeddings, which I believe will further increase the impact of this work. I would like to see this paper accepted and confirm my score.


Review 4

Summary and Contributions: The paper explains an approach for embedding high dimensional bayesian optimization problems that circunvents several issues found before.

Strengths: Explainability Clarity Significance

Weaknesses: Novelty Theoretical work

Correctness: Yes I think

Clarity: Yes, it is excellently written, it a different style that is very didactic and easy to follow.

Relation to Prior Work: Yes, and it examines several issues found before proposing solutions.

Reproducibility: Yes

Additional Feedback: Author rebuttal: I have read the author's response and talk with the other reviewers about this paper. I keep my score into 7, I think this is a good paper. ========= Good paper exposing several issues found in High BO that is given along with its code to ensure reproducibility. I have found your work very interesting. Although, I have some doubts that must be answered if more grade wants to be achieven. For future work I suggest the extension of this work to multi-objective BO. Concerning the constraints introduced by Eq 1, would the optimum still be accesible in that regions? Can it be proven theoretically? I found this aspect the most critical thing of the methodology presented in this paper. I have read the probability section of finding the optimum but I still miss a clear comparative of this probability wrt previous approaches and if, in practice, is high or low. Is proposition 1 valid for any stationary kernel or it this entire result constrained to the ARD kernel substituting the euclidean distance for a Mahalanobis one? It would be great to generalize to any kernel to give the method more flexibility. Why is TuRBO not tested in the CIFAR10 experiment?

[Author Response · NeurIPS 2020]

We thank the reviewers for their detailed reviews and constructive feedback. Below we respond to each review as much
as space allows, to provide clarification on points of confusion and answer the questions raised. We are grateful to see
that the responses are unanimously positive and we hope this work will be beneficial to the field as a whole.

**Reviewer #1**

*Improvement of ALEBO*: Thanks for raising this. On stationary problems with low-d structure, the magnitude of
improvement is large (Fig. 5). Now, as we show, real-world problems can be more complicated and while the
improvement over REMBO remained large, local-search methods were highly competitive. However, there are still
settings where linear embedding BO (and thus ALEBO) would be the best choice. The promise of embedding BO is
that all of the BO machinery developed over the years can be applied directly to HDBO. For instance, BO can maintain
high sample efficiency with high parallelism (e.g., 100 total iterations spread across 25 workers, where iterations take
hours or days). The same is not true for local search methods, including TuRBO, which requires sequential iterations
to move the trust region. Other settings where BO is not matched by local search include cost-aware, multi-task, and
multi-fidelity, to name a few. We will add discussion of this in the extra page.

*Selecting $d_e$*: This is a great point we will discuss in more detail. In some problems (e.g. robot locomotion) there is
domain knowledge. Practically, the evaluation budget will be an important factor: 500 function evaluations will support
a higher embedding dimension than 25. Sensitivity is explored in S9, and ALEBO is shown to be better than prior work.

*Constraint on NASBench*: See R3. *Supplemental*: Thanks for the suggestion, we will update to improve clarity!

**Reviewer #2**

*Clarifications*: Thanks for pointing these out, we will clarify them. k=4 is the recommendation made in the REMBO
paper, which does some sensitivity analysis. L213: the random subspace will not be axis aligned w.p. 1.

*Selecting $d_e$*: This was also brought up by R1 and is clearly a topic of importance, which has not been thoroughly
explored by the embedding BO literature. The Mahalanobis kernel can be sample-efficient despite the quadratic number
of hyperparameters parameters because of the posterior sampling, which avoids overfitting (Fig. S2). The optimization
in Fig. 5(center) used $d_e$=12, yet had excellent performance already at 25 iterations. We will add discussion of this.

*Kernel evaluation*: Prop. 1 gives a generative model for the kernel starting from a d-dim ARD RBF. We will add the
requested comparison; from the theoretical result in Prop. 1 there is little reason to doubt its performance.

**Reviewer #3**

*On performance*: Thanks for the review, we agree that one conclusion of the paper is that linear embedding BO is not
appropriate in every case. But we do want to highlight that there are other reasons why one might still favor linear
embedding BO over methods like local search (CMA-ES, TuRBO) that performed strongly in our results; see the
response to R1, which describes high parallelism and multi-fidelity optimization as two such settings.

*NASBench*: Real problems of interest to us have constraints, and CMA-ES and TuRBO do not guarantee constraint
satisfaction. We added results where we apply them via low objective for infeasibility, and ALEBO remained best.

*Constrained BO*: The biggest benefit of linear embedding BO is the ability to directly apply existing BO techniques. In
Fig. 6, we actually did use the constrained EI of Gardner et al. 2014; this is described in Sec. S5. Random embeddings
are especially useful for constrained BO because we can maintain the same embedding for all outcomes. The method is
agnostic to the acquisition function, and cPES or cMVES could be used just as easily. We'll move this to the main text.

*Nonlinear embeddings*: Thanks for the suggestion, we will add discussion of this in the extra page. In short, the main
findings all apply to the nonlinear case. A GP must be able to fit well in the embedding. End-to-end training a VAE to
include GP likelihood is an important first step, but then the same considerations apply for handling box bounds and
maintaining optima in the embedding. We will discuss potential extensions of our solutions.

**Reviewer #4**

*MOO*: Thanks for the suggestion. As discussed above, a benefit of linear embedding BO is that techniques like MOO
can be directly applied. Similar to how constraints are handled in Sec. S5, we would evaluate multiple objectives in the
embedding and use a MOO acquisition function. We will add discussion of this.

*Popt*: Thanks for raising this, we can increase clarity around this. The constrained space is not guaranteed to contain an
optimum; this is the Popt evaluated in Sec. 5. Under the problem prior used there, Popt for ALEBO is higher than for
HeSBO. REMBO can use a larger space by clipping to the boundaries, but this makes the function harder to model, and
so it is harder to find the optimum even if it is in the space.

*Prop 1*: The Mahalanobis kernel is specific for ARD RBF, but the corresponding result for a stationary kernel is that
stationary in the true space implies stationary in the embedding (a result that does not hold with clipping to box bounds).
*CIFAR*: See R3; it will be added.

[Meta-Review · NeurIPS 2020]

The paper has been actively discussed after the rebuttal that the reviewers found useful and actionable (e.g., about the practical usage of the method---choice of its hyperparameters and overall competitiveness---and about the novelty/Incremental aspects of the submission). The paper is recommended for acceptance. All reviewers have acknowledged that the paper makes a step towards better understanding BO in higher-dimensional problems. The paper (with ALEBO) should also set a much better baseline than REMBO for future research. As promised in the rebuttal, it is important to include in the final version of the paper the new elements such as (i) a discussion about the flexibility and (ii) results for TurBO/CMA-ES.